Effects of guanylurea, the transformation product of the antidiabetic drug metformin, on the health of brown trout (Salmo trutta f. fario)

Jacob Stefanie stefanie.jacob@uni-tuebingen.de 1
Knoll Sarah 2
Huhn Carolin 2
Köhler Heinz-R. 1
Tisler Selina 3
Zwiener Christian 3
Triebskorn Rita 1 4
1 University of Tübingen, Animal Physiological Ecology , Tübingen , Germany
2 University of Tübingen, Effect-based Environmental Analysis , Tübingen , Germany
3 University of Tübingen, Environmental Analytical Chemistry , Tübingen , Germany
4 Steinbeis Transfer Center for Ecotoxicology and Ecophysiology , Rottenburg , Germany
Anderson Todd
Electronic publication date: 2019 Jul 11
Publication date: 2019
Volume: 7
Electronic Location ID: e7289
Received 2019 Apr 30; Accepted 2019 Jun 12
Copyright: ©2019 Jacob et al.
Copyright year: 2019
Copyright holder: Jacob et al.
License: This is an open access article distributed under the terms of the Creative Commons Attribution License, which permits unrestricted use, distribution, reproduction and adaptation in any medium and for any purpose provided that it is properly attributed. For attribution, the original author(s), title, publication source (PeerJ) and either DOI or URL of the article must be cited.
License URL: https://creativecommons.org/licenses/by/4.0/

Keywords: Transformation product, Pharmaceutical, Brown trout, Histopathology, Lipid peroxides, Stress proteins, Swimming behaviour

Funding: Ministry for Science, Research and Arts of Baden- Württemberg 33-5733-25-11t32/2 German Excellence Initiative commissioned by the German Research Foundation (DFG) This study is funded by the Ministry for Science, Research and Arts of Baden- Württemberg (Grant No. 33-5733-25-11t32/2). Minor parts of the project were financed by the German Excellence Initiative commissioned by the German Research Foundation (DFG). The funders had no role in study design, data collection and analysis, decision to publish, or preparation of the manuscript.

==============================
Background

Guanylurea is the main transformation product of the antidiabetic drug metformin, which is one of the most prescribed pharmaceuticals worldwide. Due to the high rate of microbial degradation of metformin in sewage treatment plants, guanylurea occurs in higher concentrations in surface waters than its parent compound and could therefore affect aquatic wildlife. In this context, data for fish are scarce up to now which made us investigate the health of brown trout (Salmo trutta f. fario) in response to guanylurea.

Methods

In two experiments, eggs plus developing larvae and juvenile brown trout were exposed to three different concentrations of guanylurea (10, 100 and 1,000 µg/L) and, as a negative control, filtered tap water without this compound. Low internal concentrations were determined. The investigated parameters were mortality, length, weight, condition factor, tissue integrity of the liver and kidney, levels of stress proteins and lipid peroxides, as well as behavioural and developmental endpoints. It was found that guanylurea did not significantly change any of these parameters in the tested concentration range.

Results

In conclusion, these results do not give rise to concern that guanylurea could negatively affect the health or the development of brown trout under field conditions. Nevertheless, more studies focusing on further parameters and other species are highly needed for a more profound environmental risk assessment of guanylurea.

Introduction

For the environmental risk assessment of chemicals, the inclusion of transformation products and the investigation of their contribution to the risk posed by the parent compound are often challenging and increase the complexity of the evaluation process. In this context, it is a major issue that the identity of transformation products is often unknown; in addition, there is a general dearth of data concerning the ecotoxicological effects of identified ones (Celiz, Tso & Aga, 2009). However, particularly in cases of transformation products with high formation yield or a higher persistence and toxicity than their parent compound, it would be a striking error to neglect transformation products in risk assessment (Escher & Fenner, 2011). For example, Schlüter-Vorberg et al. (2015) showed that carboxy-acyclovir, a transformation product of the antiviral drug acyclovir formed in sewage treatment plants, was much more toxic than its parent compound. Valsartanic acid, the transformation product of the antihypertensive drug valsartan formed in the activated sludge of waste water treatment plants, was shown to be much more persistent than its parent compound (Berkner & Thierbach, 2014; Helbling et al., 2010). An example for a transformation product with a high formation yield is guanylurea. The substance is generated in sewage treatment plants by the microbial degradation of the antidiabetic drug metformin (Scheurer et al., 2012; Tisler & Zwiener, 2018; Trautwein & Kümmerer, 2011) and is detected in surface waters at concentrations up to 28 µg/L (Scheurer et al., 2012); the maximum concentration of the parent compound is about a factor of 10 lower (Scheurer et al., 2012). Generally, data on the ecotoxicity of guanylurea are scarce (Riegraf et al., 2017). To the best of our knowledge, there is only a single study addressing the effects of guanylurea in fish. In this study, larval Japanese medaka (Oryzias latipes) were exposed test to 1–100 ng/L guanylurea for 28 days in an early life stage resulting in significantly reduced growth at all tested concentrations (Ussery et al., 2019). Mortality, time to hatch, and hatching success were not influenced by the chemical. In the same study by Ussery et al. (2019), growth was not significantly reduced in adult medaka exposed to 1 ng/L and 7.5 µg/L guanylurea in a full life cycle test. In addition, the authors reported that the hormonal system seemed to be influenced by guanylurea, since enhanced estradiol production was measured in the liver of male adult medaka exposed to 7.5 µg/L guanylurea. There are few studies dealing with the effects of guanylurea in invertebrates. In crustaceans, a low sensitivity to guanylurea was shown: Daphnia magna was immobilised with a 50% effective concentration (EC50) value of 40 mg/L (Markiewicz et al., 2017). In another study, reproduction of Ceriodaphnia dubia remained unaffected with a no observed effect concentration (NOEC) of 8 mg/L (Caldwell et al., 2019).

In our study, we investigated whether and how guanylurea influences the development and health of brown trout (Salmo trutta f. fario), a species of high environmental relevance for Central Europe, at several organisational levels. Studies were conducted with larvae developed from exposed eggs and nine month-old juvenile fish exposed for three weeks to cover sensitive life stages of the selected species. The investigated parameters were apical endpoints as mortality, body weight and length, condition factor, as well as the time to hatch and heart rate of the larvae to account for embryotoxicity. Moreover, we investigated the tissue integrity of the most important organs in metabolism and biotransformation, i.e., the liver and kidney, since histopathology is a good marker to detect sublethal effects of chemicals on cells and organs (Bernet et al., 1999; Johnson et al., 1993; Schwaiger et al., 1992; Triebskorn et al., 2007). As general, biochemical stress markers, the level of the 70 kDa heat shock protein family (Hsp70) was analysed in the head of juvenile brown trout and the lipid peroxide level was measured in the head of larval brown trout. Hsp70 induction is part of the molecular stress response which is highly conserved across a wide range of taxa (Margulis, Antropova & Kharazova, 1989). Their function lies in the folding of nascent proteins and the re-folding of partially damaged proteins. Lipid peroxides are the product of reactions of reactive oxygen species (ROS) with polyunsaturated lipids (Monserrat et al., 2003). ROS are generated by different biochemical processes, e.g., by aerobic respiration or the metabolism of xenobiotics (Betteridge, 2000; Valavanidis et al., 2006). In case of an imbalance between the ROS production and the anti-oxidative defence, oxidative stress can arise (Betteridge, 2000). For example, alterations of the membrane fluidity in mitochondria caused by lipid peroxides (Chen & Yu, 1994) can lead to a loss of essential cellular functions (Hermes-Lima, Willmore & Storey, 1995). The parent compound of guanylurea, the antidiabetic drug metformin, was shown to decrease the stress protein level (Piro et al., 2012). Concerning the effects of the pharmaceutical on oxidative stress, the literature states that metformin is able to reduce (Anurag & Anuradha, 2002; Bonnefont-Rousselot et al., 2003) but also increase (Dai et al., 2014; Lee et al., 2019) the level of oxidative stress. Therefore, we wanted to investigate whether guanylurea has similar effects on Hsp70 and lipid peroxide levels as its parent compound. Additionally, for the juvenile brown trout, swimming behaviour (total distance moved and mean velocity) was analysed, since physiological stress may manifest itself as behavioural alterations (Scott & Sloman, 2004). The parent compound metformin is known to influence the aggression behaviour of Siamese fighting fish (Betta splendens). Thus, we wanted to know whether guanylurea is able to change the movement activity of brown trout. Finally, the uptake of guanylurea into the tissue of the brown trout larvae was determined.

Material and Methods

Test organisms

Brown trout (Salmo trutta f. fario) were purchased from a commercial fish breeder (trout breeding Lohmühle, D- 72275 Alpirsbach-Ehlenbogen) whose fish breeding is listed as category I, disease-free according to the EC Council Directive (European Union, 2006). Eggs in the eyed-ova stage (46 days post fertilisation (dpf)) were exposed directly after purchase. Juvenile fish (age: nine month) were kept in a 250 L aquarium in the laboratory for two weeks prior to the exposure experiments for acclimation to the lab conditions.

Test substances

Guanylurea sulphate (CAS number: 591-01-5; Lot: WIA7F and AKJLG; purity: 98%) was purchased from Tokyo Chemical Industry (Tokyo, Japan). With a maximum solubility in water of 50 g/L (ChemIDplus, 2018), the substance was readily soluble in water. The concentrations of guanylurea investigated in the present study refer to the free base.

Exposure experiments and sampling

Larval brown trout

Eyed eggs of brown trout (46 dpf) were exposed to four different nominal concentrations of guanylurea (0, 10, 100 and 1,000 µg/L) in triplicate at 7 °C in a climate chamber (start: 29.12.2017). The exposure was conducted in a semi-static system using glass aquaria with 30 test organisms per 10 L test medium. After 91 days of exposure, the water volume was increased to 15 L per aquarium. In total, 360 individuals were investigated for their development and health. During the experiment, hatching and mortality were recorded daily. At day 40, when the larvae reached the finfold resorption phase (step 38/39 according to Killeen, McLay & Johnston (1999)), the heart rate was determined. Fifteen larvae (five per replicate aquarium) from both the control and the 1,000 µg/L guanylurea treatment were transferred to vessels containing the respective test medium and the heart rate of each individual was counted for 20 s. Twice a week, 50% of the exposure medium was exchanged with freshly prepared medium. Aerated and filtered tap water (iron filter, active charcoal filter, particle filter) was used for the preparation of the medium. The medium in the aquaria was aerated with air stones (JBL Pro Silent Aeras Micro S2). The illumination conditions were kept constant during the test with a 10 h/14 h –light/dark cycle. Additionally, the aquaria were shaded from direct light using black plastic foil. After yolk-sac consumption (day 56), the larvae were fed every day with commercial trout food (INICIO plus 0.5 mm from Biomar, Denmark). The amount of food provided per day was constantly adapted in relation to the developmental status of the brown trout; the larvae were fed at maintenance. During the water exchange process, excess food and faeces were removed. Temperature, pH, oxygen content and conductivity were monitored at the beginning and end of the test as well as at day 55 (temperature = 7.3 ± 0.2 °C, pH = 8.33 ± 0.05, oxygen concentration = 11.00 ± 0.20 mg/L, conductivity= 483 ± 15 µS/cm). Eight weeks after yolk-sac consumption (17.04.2018; after 110 days) the fish were euthanised with an overdose of MS 222 (1 g/L buffered by NaHCO3) and subsequent severance of the spine. The length and weight of the fish as well as possible abnormalities or injuries were recorded. Fulton’s condition factor was calculated (ratio of weight and length cubed). Due to the small size of the test organisms, individual fish were separated into three groups (samples obtained from 10 individuals per replicate aquarium; 30 individuals per treatment). In the first group, the liver was sampled for histopathological examination. The second group provided samples for the analyses of lipid peroxide level (head). In the third group, samples for chemical analysis of fish tissue were obtained; for this purpose, the middle part of the fish was used (tissue between neck and dorsal fin). Samples for histological analysis were chemically fixed in glutardialdehyde. All other samples were immediately frozen in liquid nitrogen and stored at −80 °C until further analysis.

Juvenile brown trout

Juvenile brown trout (age: nine month) were exposed to four different nominal concentrations of guanylurea (0, 10, 100 and 1,000 µg/L) in triplicate at 7 °C in a climate chamber. The exposure was conducted in a semi-static system using glass aquaria with ten test organisms per 15 L test medium. In total, 120 individuals were investigated. The water exchange and the illumination conditions were the same as described above. Daily, juvenile brown trout were fed a defined amount of commercial trout food (INICIO plus 0.8 mm from Biomar, Denmark). Temperature, pH, oxygen content and conductivity of the media in the aquaria were monitored at the beginning and end of the test (temperature = 6.9 ± 0.3 °C, pH = 8.33 ± 0.12, oxygen concentration=11.42 ± 0.15 mg/L, conductivity = 497 ± 23 µS/cm). After 29 days of exposure (08.08.2017–05.09.2017), seven fish per aquarium (21 per treatment) were euthanised with an overdose of MS 222 (1 g/L buffered by NaHCO3) and subsequent severance of the spine. The length and weight of fish, as well as possible abnormalities or injuries, were recorded. The liver and kidney were sampled for histopathological examination. The head was taken for the analysis of the stress protein level. Samples for histological analyses were chemically fixed in glutardialdehyde. All other samples were immediately frozen in liquid nitrogen and stored at −80 °C. The remaining three fish per aquarium (nine per treatment) were used for swimming behaviour experiments (07.09.2017). In addition to the negative control in the lab, 20 juvenile brown trout were sampled directly at the trout breeding facility as a qualitative ‘hatchery control’ (12.09.2017).

Chemical analyses

During the experiments, water samples were taken to determine the real guanylurea concentrations in the test medium. For both experiments, water samples were taken at the beginning (larvae: 29.12.2017; juveniles: 08.08.2017) and end of the experiment (larvae: 17.04.2018; juveniles: 05.09.2017). Moreover, samples were taken during the exposure, before and after water exchange (larvae: after 25, 56 and 77 days; juveniles: after 14 days). The samples were stored at −20 °C until further processing. At the end of the experiment, tissue samples of the brown trout larvae (between neck and dorsal fin) containing liver, muscle, gut and kidney were investigated to determine the internal guanylurea concentration in the fish.

Analysis of water concentrations by LC-MS

The real water concentrations were determined using LC-MS with a 1290 Infinity HPLC system (Agilent Technologies, Waldbronn, Germany) and a quadrupole time of flight mass spectrometer (6550 iFunnel QTOF; Agilent Technologies, Santa Clara, CA, USA). For separation, a Phenomenex LUNA 5 u HILIC 200 A column (150 × 3 mm; 5 µm particle size) with a flow rate of 0.5 mL/min at 40 °C was used. A gradient elution was performed with eluent A (aqueous buffer with 15 mM ammonium formate and 0.1% formic acid) and eluent B (acetonitrile with 0.1% formic acid) (all chemicals purchased from Fisher Scientific, Schwerte, Germany). A portion of 95% of eluent B was used for 0-4 min, decreased to 50% within 4 min and held for 6 min. After switching back to the starting conditions, the post time was 8 min.

Samples were kept in the autosampler at 10 °C. The injection volume was 20 µL. All samples had a composition of 90% acetonitrile and 10% H2O due to dilution with acetonitrile. The ionisation of guanylurea was performed in the positive ionisation mode. Further details on operating parameters of the QTOF are provided in Paragraph S1 and Table S1. Acquired data were processed with the software Mass Hunter (Agilent Technologies). For quantification and confirmation, the exact mass of guanylurea [M+H]+ m/z 103.0614 ± 10 ppm was used at a retention time of 4.87 ± 0.5 min. The limit of quantification (LoQ) was 300 ng/L. Intra- and interday variations of the analytical method (n = 4) were 4.6 and 8.5%.

Analysis of tissue samples by SPE and CE-MS

The guanylurea concentration in the tissue of brown trout fry was determined by capillary electrophoresis–mass spectrometry (CE–MS). For sample clean-up and extraction of guanylurea solid-phase extraction (SPE) was used. Fish samples originating from all exposure concentrations were analysed. For each exposure group, tissue samples of three individuals per replicate (nine per treatment) were pooled to reach the required detection limits. For sample preparation, frozen (−20 °C) samples were first homogenised by grinding using a mortar and pestle under liquid nitrogen. A total of 100 mg of the homogenised sample was transferred to an Eppendorf tube and 1.5 mL water was added, followed by vortexing for 30 s. After centrifugation for 15 min, the supernatant was transferred into an Eppendorf tube and was then ready for SPE extraction. Prior to the extraction, the material was conditioned with 3 × 3 mL methanol followed by 3 × 3 mL of water (LC-MS grade). After equilibrating, 1 mL of sample extract was loaded onto the cartridges. The elution of guanylurea was performed with 1 mL of a methanol/acetonitrile mixture containing 2% formic acid. The eluate was evaporated to dryness under a gentle stream of nitrogen and the concentrated residue was redissolved in 300 µL methanol. After filtration using a 45 µm PTFE filter (pore size 0.45 µm, Chromafil, Macherey-Nagel, Germany), the samples were analysed by CE–MS. Calibration was performed between 5 and 50 µg/L in fish extract and the recovery was determined to 84%. The intraday variation of the analytical method was 9%. Further details can be found in the Supplemental Information.

All analyses were performed using an Agilent CE 7100 interfaced to an Agilent 6550 iFunnel Q-TOF mass spectrometer (Agilent Technologies, Waldbronn, Germany and Santa Clara, CA, USA) using an electrospray ionisation source assisted by the sheath-liquid interface. CE separations were carried out by means of an uncoated fused-silica capillary (length 80 cm, i.d. 50 µm). The background electrolyte was a mixture of 25 mM ammonium acetate and 3% glacial acetic acid in methanol. Samples were injected hydrodynamically by applying a pressure of 100 mbar for 10 s. The CE capillary was kept at 25 °C during CE runs and a voltage of + 30 kV was applied. Details of the CE–MS method are given in Paragraph S2.

Histopathological investigations

Samples for histological analyses were fixed in 2% glutardialdehyde (Merck, Darmstadt, Germany) diluted with a sodium-cacodylate buffer (0.1 M, pH 7.6; AppliChem, Darmstadt, Germany). After being washed three times with this buffer, samples were dehydrated in a graded series of ethanol and infiltrated with paraffin wax (Parablast; Leica, Wetzlar, Germany) in a tissue processor (Leica TP 1020). Additionally, kidney samples were decalcified prior to the infiltration step using a 1:2 mixture of formic acid and 70% ethanol. After paraffin embedding, samples were cut into 3 µm slices using a microtome (Leica SM 2000 R). One portion of the slices was stained with hematoxylin-eosin (to visualise nuclei, cytoplasm, connective tissue and muscles), the other one was stained with alcian blue-PAS (to visualise mucus and glycogen). The slides were examined using a light microscope (Axioskop 2; Zeiss, Oberkochen, Germany), first in a qualitative manner to obtain a general overview of the tissues and to identify pathologies. In a second step, the observed pathologies were semi-quantitatively assessed and classified into one of five different categories (1: control, 2: slight reaction, 3: moderate reaction, 4: strong reaction, 5: destruction) according to the criteria published by Triebskorn et al. (2008). In a further step, all samples were re-analysed once again after being blinded and randomised to avoid observer bias in the final evaluation.

Stress protein analysis

For the determination of the 70 kDa stress protein family (Hsp70) level in the heads of juvenile brown trout (21 per treatment), the samples were homogenised with a mixture of 98% extraction buffer and 2% protease inhibitor as described by Dieterich et al. (2015). Subsequently, the total protein content in the samples was quantified according to Bradford (1976). A standardised amount of 40 µg total protein per sample was used for the analysis of the Hsp70 level. The proteins were separated according to their weight using minigel SDS-PAGE (sodium dodecyl sulphate polyacrylamide gel electrophoresis) and blotted on a nitrocellulose membrane in a semi-dry chamber. After specific binding of the primary antibody (monoclonal α-Hsp70 IgG; Dianova Hamburg, Germany) to the Hsp70 proteins, a secondary antibody (peroxidase-coupled α-IgG; Jackson Immunoresearch, West Grove, PA, USA) directed to the first antibody was added. Finally, the membranes were stained with 4-chloro-1-naphthol until the Hsp70 proteins became visibly. The optical volume (=band area × average grey scale value) of the bands was quantified in relation to an internal standard (brown trout full body homogenate).

Lipid peroxide analysis

The level of lipid peroxides in the head of the brown trout larvae (30 per treatment) was determined with the FOX (ferrous oxidation xylenol orange) assay according to a modified version of the protocols from Hermes-Lima, Willmore & Storey (1995) and Monserrat et al. (2003). Basically, Fe(+II) is oxidised by the lipid peroxides of the sample under acidic conditions, followed by a complexation of the resulting Fe(+III) with the dye xylenol orange resulting in a colour change. Head samples were homogenised in HPLC-grade methanol (according to a tissue: methanol ratio of 1:7, w.w.) and centrifuged at 4 °C at 14,000 rpm for 5 min. 50 µL each of FeSO4, H2SO4 and xylenol orange, 35 µL sample supernatant and 15 µL bi-distilled water were pipetted in each well, adding up to a total volume of 200 µL. Each sample was tested in triplicate and, additionally, tested without FeSO4 to account for endogenous iron in the sample. Subsequently, the samples were incubated for 90 min at room temperature. Then, the absorbance was measured at 570 nm using an automated microplate reader (Bio-Tek Instruments, Winooski, VT, USA). Finally, 1 µL of 1 mM cumene hydroperoxide (CHP) solution was added per well and the samples were incubated for 30 min at room temperature. Then, the absorbance was measured again and cumene hydroperoxide equivalents (CHPequiv.) were calculated according to the following equation: CHPequiv.=Abs570beforeCHPAbs570afterCHP∗volumeCHP∗totalvolumesamplevolume∗dilutionfactor

with volume CHP = 1 µL; total volume = 200 µL; sample volume = 25 µL; dilution factor = 4.

Analysis of swimming behaviour

The swimming behaviour of the juvenile brown trout was quantitatively analysed using the EthoVision 12 software (Noldus, Wageningen, the Netherlands). Three fish per replicate (nine per treatment) were transferred to cubic aquaria (17 cm edge length) containing 1.6 L of the respective test medium. After acclimation for 2 min, swimming behaviour was recorded for 18 min with a digital camera (Basler acA1300-60 gm camera, 1.3 MP resolution; Basler, Ahrensburg, Germany) in four aquaria simultaneously. By tracking the centre-point of the fish, the total distance moved and the mean velocity of each individual were analysed. Afterwards, possible identity swaps between the tracked individuals were corrected with the Track Editor in EthoVision software.

Statistical analysis

Statistics were conducted with JMP 12 (SAS, Cary, NC, USA). Whenever normal distribution and homoscedasticity were not present, data were transformed. The data of body weight of larval brown trout and the data of the swimming behaviour were transformed with a natural logarithm function. Body length and weight of the juveniles as well as the data of the lipid peroxide analysis of the larvae were transformed with the function x−0.5. Mortality and time to hatch were analysed with Cox-regression. The histopathological data were checked for significance with the likelihood-ratio-χ2-test. For the analysis of the lipid peroxide level of the larvae, a Welch ANOVA was performed, since the data could not be transformed to reach homoscedasticity. A nested ANOVA was conducted for all other endpoints using the replicate as a nesting factor. The α-level was set to 0.05, but was adjusted in case of multiple testing according to sequential Bonferroni correction. Comparisons with the hatchery controls were solely conducted qualitatively. The statistical tests used and the corresponding p-values are shown in the Results section; further information (e.g., degrees of freedom and F-values) is given in Paragraph S5.

Animal welfare

All experiments were approved by the animal welfare committee of the Regional Council of Tübingen, Germany (authorisation ZO 2/16).

Credibility of the data

The details about the fulfilment of the criteria for reporting and evaluation ecotoxicity data (CRED) according to Moermond et al. (2016) are provided in Paragraph S6.

Results

Chemical analyses

The measured guanylurea concentrations in exposure media were in good accordance with the nominal water concentrations (Tables 1 and 2, detailed information in Paragraph S1, Tables S2 and S3). Therefore, we refer to the nominal concentrations throughout the entire text of this paper. In most exposure groups, the real concentrations were slightly higher than the nominal concentrations; the recovery rate was between 89 and 132%. The analysis of the tissue of brown trout larvae revealed that guanylurea was solely quantifiable in the exposure group with the highest guanylurea concentration (1,000 µg/L) and only at low concentrations. In the three replicates of this exposure group, the transformations product of metformin was found at a concentration of 5.05 ng/g in the first replicate, 16.91 ng/g in the second replicate and 9.98 ng/g in the third replicate, resulting in an average internal guanylurea concentration of 10.65 ng/g (Table 1).

Table 1 Measured guanylurea concentrations in medium and in tissue, mortality, biometric data, developmental and biochemical parameters of larval brown trout exposed to guanylurea.

All data are displayed as arithmetic means ± standard deviations. The heart rate was only evaluated for the negative control and the highest guanylurea concentration.

Larval brown trout			
Nominal water concentrations [µg/L]	0	10	100	1000	
Measured (real) water concentrations	<LoQ	9.0 ± 0.9 µg/L	117.3 ± 5.3 µg/L	1254 ± 33 µg/L	
Internal concentration (average for 3 replicates) [ng/g]	<LoQ	<LoQ	<LoQ	10.65	
Mortality [%]	5.6 ± 1.6	2.2 ± 3.1	6.7 ± 2.7	3.3 ± 0.0	
Time to hatch (dpf)	55.5 ± 3.0	56.8 ± 1.6	57.0 ± 1.1	56.2 ± 2.1	
Heart rate [bpm]	59.2 ± 5.1	n.e.	n.e.	59.4 ± 4.3	
Body weight [g]	0.50 ± 0.11	0.50 ± 0.14	0.51 ± 0.13	0.50 ± 0.12	
Body length [cm]	3.8 ± 0.3	3.7 ± 0.3	3.8 ± 0.3	3.7 ± 0.3	
Condition factor [g/cm3]	0.92 ± 0.10	0.94 ± 0.09	0.92 ± 0.11	0.99 ± 0.11	
Lipid peroxide level [CHP-equiv./mg wet weight]	18.01 ± 5.57	16.97 ± 4.61	18.55 ± 7.14	19.48 ± 5.24	
Notes.

LoQ limit of quantification

n.e. not evaluated

bpm beats per minute

dpf days post fertilisation

Table 2 Measured guanylurea concentrations in medium, mortality, biometric data, biochemical and behavioural parameters of juvenile brown trout exposed to guanylurea. All data are displayed as arithmetic means ± standard deviations.

Juvenile brown trout		
Nominal water concentrations [µg/L]	0	10	100	1000	Hatchery control	
Measured (real) water concentrations [µg/L]	<LoQ	11.8 ± 0.2	132.5 ± 2.4	1319 ± 60	-	
Mortality [%]	3.3 ± 4.7	3.3 ± 4.7	0.0 ± 0.0	0.0 ± 0.0	-	
Body weight [g]	2.80 ± 1.01	2.88 ± 0.83	2.83 ± 0.88	3.06 ± 0.80	8.66 ± 2.64	
Body length [cm]	6.4 ± 0.7	6.5 ± 0.7	6.5 ± 0.7	6.6 ± 0.6	9.1 ± 0.8	
Hsp 70 level [relative grey value]	1.15 ± 0.18	1.10 ± 0.20	1.18 ± 0.18	1.08 ± 0.21	1.09 ± 0.07	
Total distance moved [cm]	781.5 ± 786.2	1432.7 ± 1016.4	972.0 ± 452.0	912.3 ± 658.6	-	
Mean velocity [cm/s]	0.7 ± 0.7	1.3 ± 0.9	0.9 ± 0.4	0.8 ± 0.6	-	
Notes.

LoQ limit of quantification

Mortality and growth metrics

The mortality varied between 2.2 and 6.7% in the experiment with the larval brown trout (Table 1) and between 0 and 3.3% in the experiment with juvenile brown trout (Table 2), so guanylurea did not show any signs of a lethal effect (COX-regression: larvae: p = 0.2001; juveniles: p = 0.4253) in the tested concentration range. Also, body weight, length and condition factor of both life stages of brown trout were not influenced by guanylurea (Table 1 and 2) (nested ANOVA: larvae: p = 0.8919 (weight); p = 0.0589 (length); p = 0.0587 (condition factor); juveniles: p = 0.6383 (weight); p = 0.8157 (length); p = 0.7942 (condition factor)).

Developmental parameters

The investigations of heart rate and time to hatch did not reveal any effects of guanylurea in larval brown trout (Table 1) (heart rate: nested ANOVA: p = 0.9090; time to hatch: COX-regression: p = 1). Moreover, the hatching success in all exposure groups was 100%.

Biochemical markers

The analysis of lipid peroxides in larval brown trout did not reveal any differences between the exposure groups (Table 1) (Welch-ANOVA: p = 0.2333). Also, the stress protein level in juvenile brown trout was not influenced by guanylurea (Table 2) (nested ANOVA: p = 0.3235).

Swimming behaviour

The total distance moved and mean velocity of juvenile brown trout exposed to 10 µg/L guanylurea were slightly, but not significantly enhanced compared to the control (Table 2). This was not observed after exposure to the higher test concentrations (nested ANOVA: distance moved: p = 0.1676; mean velocity: p = 0.1676).

Histopathology

Liver

The liver tissue of larval brown trout did not show any severe damage or pathological alterations and was classified into the categories 1, 2 or 3. For the liver samples of juvenile brown trout, categories 1 to 4 were allocated to the sections. In both life stages, one portion of the liver samples showed large cells with a bright cytoplasm (Figs. 1A, 1C) containing high amounts of glycogen (Fig. 1B). The other portion of the samples revealed small and darker hepatocytes (Fig. 1D) and a low glycogen amount (Fig. 1E). The glycogen amount was generally lower in the livers of juveniles compared to larval brown trout. Moreover, large inflammatory sites (Fig. 1F), which occurred in some livers of juvenile brown trout exposed to guanylurea, resulted in a classification into category 4. Further details of the qualitative tissue analyses for each individual are given in Paragraph S4. The semi-quantitative histopathological assessment of the liver did not show any significant effects of guanylurea in larval or juvenile brown trout (Likelihood-ratio- χ2-test: larvae: p = 0.2865; juveniles: p = 0.5224) (Fig. 2). In the hatchery control, the livers contained more glycogen compared to the juvenile brown trout exposed in laboratory, but the prevalence of inflammatory sites resulted in a classification of the respective samples into categories 3 and 4 (Fig. 2).

Figure 1 Representative sections of liver and kidney of brown trout.

Control status of liver of (A) juvenile and (C) larval brown trout with large hepatocytes with bright cytoplasm and (B) high glycogen amounts; reaction status of liver with (D) small hepatocytes with (E) reduced glycogen amounts and (F) inflammatory sites; control status of kidney showing (G) proximal tubules in compact hematopoietic tissue; reaction status of kidney showing (H) hyaline droplets and (I) vacuoles in the proximal tubules; (B & E): alcian blue-PAS staining; all other sections: haematoxylin-eosin staining; (C): liver section of larval brown trout; all other sections: juvenile brown trout.

Figure 2 Semi-quantitative assessment of histopathological symptoms in the (A) liver of larval brown trout and in the (B) liver and (C) kidney of juvenile brown trout exposed to guanylurea.

Samples were classified in categories 1 to 4; category 5 was not allocated to any of the samples. The number n of examined individuals is given at the base of each bar. Statistical comparisons revealed that guanylurea did not significantly affect the integrity of the investigated tissues. Hatchery controls (HC) were excluded from statistical comparison.

Kidney

The kidney tissue of juvenile brown trout was classified into the categories 1 to 4. Symptoms were the enlargement of vesicles and the accumulation of hyaline droplets (Fig. 1H) in anterior parts of the proximal tubules and the occurrence of vacuoles (Fig. 1I) in anterior and posterior parts of the proximal tubules. In a few kidney sections, the hematopoietic tissue was partially degenerated and the spaces between the glomeruli and Bowman’s capsule were enlarged. The semi-quantitative assessment revealed that these symptoms were prevalent in all treatments and that there was no observable difference between the exposure groups with guanylurea and the control (Likelihood-ratio- χ2-test: p = 0.6668; Fig. 2). In the hatchery control (Fig. 1G), however, symptoms as described above did not occur except for one single animal. Further details of the qualitative tissue analyses for each individual are given in Paragraph S4.

Discussion

In the present study, we investigated the effects of guanylurea, the major transformation product of metformin, in brown trout. Generally, it is assumed that most metabolites or transformation products show the same behaviour in the environment as their parent compounds (EMA, 2006) and are equally or less toxic than these (Escher & Fenner, 2011; Kümmerer, 2009). With respect to guanylurea, however, there is a lack of ecotoxicological data, and it is not known whether this transformation product resembles metformin in its (ecotoxic) effects. Usually, a similar chemical structure of transformation product and parent compound indicates a similar environmental behaviour. The general chemical properties of guanylurea are similar to those of the parent compound: both substances are very polar and positively charged at the pH values of natural waters (Scheurer et al., 2012; Scheurer, Sacher & Brauch, 2009). It has been shown that metformin can be taken up by fish to some extend leading to tissue concentrations of about 55 ng/g in brown trout larvae exposed to 1,000 µg/L metformin (Jacob et al., 2018; Ussery, 2018). The mean guanylurea concentration in the tissue of larval brown trout exposed to 1,000 µg/L was about 11 ng/g, which is a factor of 5 lower compared to metformin. These low internal concentrations of metformin and its transformation product in fish might be related to the high polarity (logD =−3.85 for guanylurea and −5.62 for metformin; calculation according to chemicalize.com) and charge (single for guanylurea and double for metformin) of these compounds.

The present study demonstrated that guanylurea did not lead to any lethal or embryotoxic effects in brown trout. Mortality and developmental parameters, like heart rate, hatching success and time to hatch were not influenced by the transformation product of metformin. Also, Ussery et al. (2019) showed that guanylurea did not affect hatching success or time to hatch in Japanese medaka exposed for 28 days in an early life stage test at concentrations of 1–100 ng/L and for 165 days in a full life cycle test to 1 ng/L and 7.5 µg/L guanylurea. Investigations of the parent compound metformin revealed comparable results (Jacob et al., 2018; Ussery et al., 2019); neither study showed a negative impact of the pharmaceutical on survival or development up to the highest concentrations applied.

The length and weight of brown trout was not affected by guanylurea, neither in larvae nor in juveniles. When interpreting these results, one has to keep in mind that the brown trout were fed at maintenance and not ad libitum. This feeding regime was chosen due to the semi-static test design, since it is necessary to provide clean water with high oxygen content for brown trout. In contrast, Ussery et al. (2019) showed the weight and length of larval medaka to be decreased by guanylurea, but it was not significantly reduced in adult medaka exposed in a full life-cycle test. Guanylurea had no influence on the condition factor of the two life stages of brown trout; this was also observed by Ussery et al. (2019) for adult medaka. Metformin, the parent compound of guanylurea, clearly reduced the growth of fish. This was shown for early life stages of Japanese medaka (Ussery et al., 2018), but also for brown trout (Jacob et al., 2018) and adult male fathead minnows in a full life cycle test (Niemuth & Klaper, 2015). This metformin-induced effect might be explained by the appetite reducing capacity of metformin, which also explains its use as a weight loss drug (Malin & Kashyap, 2014). Guanylurea, in contrast, does not seem to show this activity in brown trout.

The biochemical analyses of lipid peroxides and stress proteins did not reveal any influences of guanylurea on these stress parameters in larval and juvenile brown trout. Likewise, in big ramshorn snails (Planorbarius corneus) exposed to 0.1–100 mg/L guanylurea for 21 days, lipid peroxides and stress proteins were not changed by guanylurea (Jacob et al., 2019). Thus, one explanation could be that guanylurea does not induce proteotoxic or oxidative stress at all. However, it could also be possible that the peroxidation of lipids was prevented by increased activity of glutathione peroxidase, superoxide dismutase or catalase, which would lead to the neutralisation of ROS. Therefore, additional analysis of these antioxidant enzymes would be of advantage to gain more detailed insights into oxidative stress exerted in fish. This, however, was not possible in the present study due to restrictions with respect to tissue availability. Further information concerning the effects of guanylurea on stress proteins and lipid peroxides—to the best of our knowledge- does not exist in the literature. The structurally similar compound guanidine serves as a scavenger for ROS, so it contributes to reducing oxidative stress (Yildiz et al., 1998). The parent compound metformin was also shown to reduce the production of ROS; its anti-oxidative effect is known for type 2 diabetes patients who are prone to oxidative stress (Bonnefont-Rousselot et al., 2003; Bułdak et al., 2014; Dehkordi et al., 2018). However, in a study with Japanese medaka, metformin increased the ROS level in male fish and the catalase activity in female fish, but it also reduced the glutathione level in male fish (Lee et al., 2019). Phenformin, an antidiabetic drug related to metformin, was shown to reduce ROS generation in the serum of rats, whereas it increased the ROS production in the brain (Anisimov et al., 2005). The same drug had no effect on the Hsp70 levels in hippocampal neurons (Lee et al., 2002). In contrast, the expression of stress protein Hsp70 was shown to be reduced by metformin in the pancreas of rats exposed to high free fatty acid levels (Piro et al., 2012). Despite all this information on chemically similar compounds, there is, currently, no indication that the transformation product guanylurea may exert or reduce oxidative or proteotoxic effects.

The swimming behaviour of juvenile brown trout also did not provide any evidence for a possible harmfulness of guanylurea. The high variation in the activity of brown trout exposed to 10 µg/L guanylurea was caused by exceptionally high swimming activity (total distance moved >2,000 cm) of three out of nine individuals in this exposure group. Unfortunately, a comparison with literature data is not possible because studies dealing with the effects of guanylurea on behaviour are lacking. As shown in a previous study, neither the parent compound metformin changed the movement activity of brown trout larvae (Jacob et al., 2018) or the locomotor activity of zebrafish (Godoy et al., 2018). However, MacLaren, Wisniewski & MacLaren (2018) showed that metformin can influence the aggressive behaviour of Siamese fighting fish (Betta splendens). Generally, endpoints of locomotor and movement activity seem to be suitable parameters for the assessment of neuroactive drugs like antidepressants (Brodin et al., 2014) as, e.g., citalopram enhanced the swimming activity of three-spine stickleback (Kellner et al., 2016). Since physiological stress effects were lacking in our experiments with guanylurea, it seems plausible that no behavioural disruption occurred.

The histopathological examinations did not reveal any guanylurea-induced changes in liver and kidney of brown trout. With respect to the two investigated life stages, the livers of larval brown trout were generally in a better condition than the livers of the juveniles. Since the test organisms in the lab were fed at maintenance but not ad libitum as the latter is the case in breeding farms such feeding regime could result in a lower hepatic glycogen amount, particularly for the larger juveniles. Symptoms like hyaline droplets and vacuoles in the proximal tubules of the kidney of the juveniles might be related to the altered water parameters, e.g., conductivity and pH, compared to the situation in the trout breeding farm. Such symptoms have also been observed in another study on brown trout performed in the laboratory (Schwarz et al., 2017). The absence of reactions in the kidney of the hatchery controls support this assumption. An increase in the hepatic glycogen amount, as it was the case for brown trout exposed to metformin (Jacob et al., 2018) indicating the therapeutic effect of the drug on glycogen depletion, could not be observed in the experiments with guanylurea.

Overall, none of the investigated endpoints in the present study revealed any effect of guanylurea on the development or health of brown trout. However, Ussery (2018) indicated that the transformation product of metformin can influence the hormone system, since the estradiol production in the liver of adult, male Japanese medaka exposed in a full life cycle test to 7.5 µg/L guanylurea was increased. This is of particular importance since the parent compound metformin was shown to exert endocrine effects in fathead minnow (Pimephales promelas) with an upregulation of vitellogenin mRNA in juveniles (Crago et al., 2016) and adult males (Niemuth et al., 2015), the occurrence of intersex in adult males and reduced fecundity of mating pairs (Niemuth & Klaper, 2015). However, vitellogenin production in the liver was not induced by guanylurea in Japanese medaka (Ussery, 2018). Therefore, future studies should focus more explicitly on this hormone-changing aspect of guanylurea and the mode of action behind it.

In general, transformation products play a minor role in the regulation of pharmaceuticals. In the European Medicines Agency (EMA) guideline on the environmental risk assessment of medical products for human use, tests according to the OECD 308 on transformation of organic chemicals in water/sediment systems are requested for the base data set of each pharmaceutical which is provided in phase II, tier A of the environmental risk assessment (EMA, 2006). These tests serve for the identification and quantification of transformation products that are formed at a level of ≥ 10% of the applied dose of the parent compound (OECD, 2002). Berkner & Thierbach (2014) examined the OECD 308 tests of dossiers for marketing authorisation applications of pharmaceuticals and found 70% of the tests to show a formation of at least one transformation product, but only 26% of the studies to identify this product. These results reveal a significant data gap regarding the transformation products of pharmaceuticals that form in the environment. To date, in the new draft for the guideline on the environmental risk assessment of pharmaceuticals of the EMA, the identification of transformation products according to the OECD 308 is no longer supposed to be part of the base data set of phase II, tier A (EMA, 2018). Consequently, the environmental risk assessment of pharmaceuticals will stay ‘blind’ for the formation and action of transformation products. In addition, an evaluation of the ecotoxic effects of the transformation products is not mandatory for an environmental risk assessment of pharmaceuticals. This, however, would be of particular importance for transformation products that are very persistent or occur at higher environmental concentrations than its parent compound, as is the case for guanylurea.

Conclusion

Our study showed that guanylurea neither led to lethal, embryotoxic nor behavioural effects in brown trout at the tested concentrations. Also, the length, weight, biochemical markers (stress proteins and lipid peroxides) and tissue integrity of the main metabolic organs (liver and kidney) were not influenced by the transformation product of metformin. Internal concentrations were very low. Investigations of ecotoxic effects of transformation products are scarce in general and this also applies to guanylurea. Thus, we consider our present study as a rather elaborate starting point, at least for a risk assessment of fish toxicity, even though we admit that a final evaluation of the ecotoxic potential of guanylurea has not been completed, particularly in view of its endocrine potential. Nevertheless, we appeal to authorities to go ahead and implement the present data into the environmental risk assessment of the parent compound metformin.

Supplemental Information

Supplemental Information 1 The chemical analysis of the exposure medium and the tissue of the test organisms, water quality parameters, histopathological examination and the statistical analysis, and information about the fulfillment of the CRED-criteria

Click here for additional data file.

This study is part of the project Effect-Net (Effect Network in Water Research) which is part of the Wassernetzwerk Baden-Württemberg and funded by the Ministry for Science, Research and Arts of Baden-Württemberg. Particular thanks go to Thomas Braunbeck, Heidelberg University, for the coordination of this project. We are grateful to Melanie Biecker and Eileen Pfitzer for the analysis of biochemical markers which were conducted in context of their bachelor theses. Furthermore, we thank Andreas Dieterich, Birgit Dittrich, Stefanie Krais, Lone Kundy, Katharina Peschke, Judith Rüschhoff, Hannah Schmieg, Simon Schwarz, Sabrina Wilhelm and Michael Ziegler for technical assistance and help in the lab.

Additional Information and Declarations

Competing Interests

Author Contributions

Animal Ethics

Data Availability

The authors declare there are no competing interests.

Stefanie Jacob performed the experiments, analyzed the data, prepared figures and/or tables, authored or reviewed drafts of the paper, approved the final draft.

Sarah Knoll and Selina Tisler analyzed the data, authored or reviewed drafts of the paper, approved the final draft.

Carolin Huhn and Christian Zwiener contributed reagents/materials/analysis tools, authored or reviewed drafts of the paper, approved the final draft.

Heinz-R. Köhler and Rita Triebskorn conceived and designed the experiments, contributed reagents/materials/analysis tools, authored or reviewed drafts of the paper, approved the final draft.

The following information was supplied relating to ethical approvals (i.e., approving body and any reference numbers):

All experiments were approved by the animal welfare committee of the Regional Council of Tübingen, Germany (authorisation ZO 2/16).

The following information was supplied regarding data availability:

The raw data for the experiments are available at the following URLs: Larval brown trout: https://effectnet-seek.bioquant.uni-heidelberg.de/studies/11, Juvenile brown trout: https://effectnet-seek.bioquant.uni-heidelberg.de/studies/12.

From the list of investigated endpoints, please select the parameter you are interested in by double-clicking on the excel-sheet and view the data by clicking on the “Download” button.

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
