# Peer review of "Effects of guanylurea, the transformation product of the antidiabetic drug metformin, on the health of brown trout (Salmo trutta f. fario)"

_PeerJ, doi:10.7717/peerj.7289_

## Round 0.1 · original submission · Major Revisions

Both reviewers have good comments that will improve your manuscript. Since those comments are clear, I won't repeat them here. Please focus your revision to address any concerns from the reviewers.

Reviewer 1 ·

Basic reporting

The authors are assessing the impact of guanylurea on the health of early life stages of brown trout. While the study’s main conclusions are a lack of effect, this information is still very important.

Introduction:
The literature references in the introduction and discussion are good, there is a sufficient amount of background provided.

Hypotheses need to be clearly established.
The hypotheses of the experiment are not stated and it is unclear why the authors picked the endpoints and life stages investigated. Please describe why those endpoints were selected and what the hypothesis is. The only endpoints that were explained briefly were Hsp70 and ROS. But even for these, it is unclear what the authors expected the effect of guanylurea to be. For example, the authors can clearly establish why they expected hsp70 to cause proteotoxic effects by linking it to previous studies. But why and how did the authors expect guanylurea to impact swimming behavior etc?

Methodology:
Overall the methodology is pretty clear but there are a few areas that need to be expanded upon. The only section that is of concern is the larval growth which may be just a misunderstanding of the reviewer.

Sample size.
Are the authors counting the tanks as the n, or the individual fish?

Feed amount
Please define how much the fish were fed as a % bodyweight per day. This is of particular importance for a growth and development study. After reading the discussion it sounds like the authors fed the fish a ration weight. Is this correct? If so the growth data should not be included as the fish should not grow under those conditions.
This study is not particularly designed with growth as an endpoint, as the fish were only sampled at the end of the study and not throughout. It would be better if the authors use the words weight and length rather than using the word growth.

Hatchery control fish.
It makes complete sense to use hatchery control fish. What is not clear is why the hatchery control fish were 2-3 times larger (body weight) than the experimental fish. Why is this, and what is the benefit or relevance of using them? Does this mean the fish were being underfed and actually lost weight over the course of the study? If so how did they also decrease in length?

Results/Discussion:
The discussion reads well. The figures/tables are appropriate.

Experimental design

This is mentioned in section 1:

-Clarify the hypotheses
-Elaborate on the methods, especially with the larval feeding. If the fish were fed at a maintenance ration, then the growth results are not relevant and should either be re-contextualized or dropped from the study.

Validity of the findings

As mentioned in the previous sections the growth aspects may not be valid. However, the rest of the findings are novel and valid. Of particular interest to this reviewer is the histopathological, stress protein, and lipid peroxide analysis.

Additional comments

The paper provides interesting information on the effects of guanylurea on a species of fish that is of relevance to European rivers. The paper presents a substantial research effort with novel data analyzing the potential effects of guanylurea. This paper could be strengthened by having a more clear direction in the endpoints measured and the expected outcomes. Some of the endpoints measured such as survival, embryotoxicity, hatch success and swimming behavior the authors did not expect to see any significant effect on. While other parameters such as hsp70, histopathology, growth, etc. the authors may have expected an outcome, but it is not clear what they were expecting to see. For histopathology, the authors mention the livers of the larval fish were in better condition than the juvenile fish due to feeding them a maintenance diet. It should be made clear why the fish were fed at maintenance, especially considering the authors were measuring growth and trying to deduce if the compound impacts growth. The authors are correct in their statement that this data is of particular importance to risk assessment, and this paper makes a valuable contribution to the understanding of guanylureas effects in fish.

Reviewer 2 ·

Basic reporting

See general comments for my review

Experimental design

See general comments for my review

Validity of the findings

See general comments for my review

Additional comments

This study examined the effects of a prolonged exposure of guanylurea on brown trout larvae and juveniles. The investigators measured internal concentrations of guanylurea, mortality, growth, movement, liver and kidney integrity, and two measures of stress, HSP70 and LPO. Results from the study indicate that there was no statistical difference between guanylurea-exposed fish and the control. The methods employed in the study were sufficient and would not affect the results, though there is some question on why some of the molecular measurements were taken (See below). Below are my concerns regarding the manuscript that I would like the reviewers to consider.

Major Concerns:

1. Why did you measure HSP70 and LPO in the head? Why not measure it in the liver? Secondly, what was the hypothesis behind these measurements? Has metformin or guanylurea been shown to illicit lipid peroxidation? I would think LPO or HSP70 response in a common pharmaceutical to be counter-productive. You discuss the possibility of increased Phase II enzymes neutralizing ROS, yet one conclusion could be that guanylurea-ROS production does not occur in the head or anywhere for that matter. Then, the authors go on to say that phenformin reduced ROS of aging rats—albeit in the head—which it appears that this reference refers to phentermine. In that study it stated that “Rats treated with this antidiabetic drug showed intermediate values of ROS generation. Differences among the groups in total antioxidant activity were not obvious.” (Ansimov et al. 2015).

2. Why wasn’t the vitellogenin data shown? That would be an interesting biomarker to compare to other fish studies.

3. Since you have length and weight, consider giving a condition factor score. Again, others have used this as a measure of fitness with exposure to metformin, so it would be a useful comparison.

4. It would be useful to have a paragraph in the Introduction on why some of these biomarkers were chosen. One could just take some of the discussion and move it to the introduction.

---

## Round 0.2 · accepted · Accept

Thank you for your efforts in revising your manuscript and addressing reviewer comments.

Reviewer 1 ·

Basic reporting

See general comments.

Experimental design

See general comments.

Validity of the findings

See general comments.

Additional comments

The authors addressed the revisions suggested by both reviewers well and the article is acceptable for publication.